# Index cases first identified by nasal-swab rapid COVID-19 tests had more transmission to household contacts than cases identified by other test types

Jenny Ji[1]☯, Alexander Viloria Winnett[1,2]☯, Natasha Shelby[1], Jessica A. Reyes[1], Noah W. Schlenker[1], Hannah Davich[1], Saharai Caldera[1], Colten Tognazzini[3], Ying-Ying Goh[3], Matt Feaster[3], Rustem F. Ismagilov[1]*

1 California Institute of Technology, Pasadena, California, United States of America, 2 University of California Los Angeles–California Institute of Technology Medical Scientist Training Program, Los Angeles, California, United States of America, 3 Pasadena Public Health Department, Pasadena, California, United States of America

☯ These authors contributed equally to this work.
* rustem.admin@caltech.edu

**Data Availability Statement:** Raw data are available at CaltechDATA: https://doi.org/10.22002/csh5w-rf132.

## Abstract

At-home rapid COVID-19 tests in the U.S. utilize nasal-swab specimens and require high viral loads to reliably give positive results. Longitudinal studies from the onset of infection have found infectious virus can present in oral specimens days before nasal. Detection and initiation of infection-control practices may therefore be delayed when nasal-swab rapid tests are used, resulting in greater transmission to contacts. We assessed whether index cases first identified by rapid nasal-swab COVID-19 tests had more transmission to household contacts than index cases who used other test types (tests with higher analytical sensitivity and/or non-nasal specimen types). In this observational cohort study, 370 individuals from 85 households with a recent COVID-19 case were screened at least daily by RT-qPCR on one or more self-collected upper-respiratory specimen types. A two-level random intercept model was used to assess the association between the infection outcome of household contacts and each covariate (household size, race/ethnicity, age, vaccination status, viral variant, infection-control practices, and whether a rapid nasal-swab test was used to initially identify the household index case). Transmission was quantified by adjusted secondary attack rates (aSAR) and adjusted odds ratios (aOR). An aSAR of 53.6% (95% CI 38.8–68.3%) was observed among households where the index case first tested positive by a rapid nasal-swab COVID-19 test, which was significantly higher than the aSAR for households where the index case utilized another test type (27.2% 95% CI 19.5–35.0%, P = 0.003 pairwise comparisons of predictive margins). We observed an aOR of 4.90 (95% CI 1.65–14.56) for transmission to household contacts when a nasal-swab rapid test was used to identify the index case, compared to other test types. Use of nasal-swab rapid COVID-19 tests for initial detection of infection and initiation of infection control may be less effective at limiting transmission to household contacts than other test types.

**Funding:** This study is based on research funded in part by the Bill & Melinda Gates Foundation (INV-023124). The findings and conclusions contained within are those of the authors and do not necessarily reflect positions or policies of the Bill & Melinda Gates Foundation. This study was also funded in part by grants from the Ronald and Maxine Linde Center for New Initiatives at the California Institute of Technology (to RFI), a grant from the Jacobs Institute for Molecular Engineering for Medicine at the California Institute of Technology (to RFI), a DGSOM Geffen Fellowship at the University of California, Los Angeles (to AVW), and the John Stauffer Charitable Trust SURF Fellowship at the California Institute of Technology (to JJ). The funders had no role in study design, data collection and analysis, decision to publish, or preparation of the manuscript. There was no additional external funding received for this study.

**Competing interests:** I have read the journal's policy and the authors of this manuscript have the following competing interests: R.F.I. is a cofounder, consultant, and a director and has stock ownership of Talis Biomedical Corp. This does not alter our adherence to PLOS ONE policies on sharing data and materials. All other co-authors report no competing interests.

## Introduction

The majority of SARS-CoV-2 transmission events occur among household contacts [1, 2]. Numerous studies have characterized household transmission of SARS-CoV-2 [3–8] and identified factors that modulate the risk of transmission within households, such as larger household size being associated with higher risk [9–12]. Similarly, disparities by race and ethnicity have been observed, while controlling for socioeconomic differences [11, 13]. Age of both the index case (first person in the household to become infected) and at-risk household contacts (who either remain uninfected or become infected secondary cases) has also been implicated in SARS-CoV-2 household-transmission patterns [6, 14–17]. Furthermore, although vaccination does not fully prevent breakthrough infections [18], vaccination has been shown to be protective and decrease the risk of infection [8, 19–23]. Specific infection-control practices, such as wearing a mask around infected contacts, physical distancing, and quarantining sick individuals have also shown protective effects [14, 19, 24–26]. Lastly, SARS-CoV-2 variants such as Delta and Omicron have been shown in large studies to have greater transmissibility compared with ancestral variants [8, 19, 20, 27–34].

Early identification of an infectious individual is a critical step to reduce subsequent transmission, including within households. Because transmission of SARS-CoV-2 occurs during both the asymptomatic and symptomatic periods of infection [35–38], diagnostic testing to quickly prompt infection control practices has been effective to limit additional exposures and transmission [39]. Conversely, infectious individuals that go unidentified or delay identification allow for greater exposure to contacts and thereby more transmission [12, 40, 41].

Delayed detection can occur due to test turnaround times or when a test yields a false-negative result. Rapid tests (e.g., antigen and some molecular tests) offer fast turnaround times, but require higher levels of virus to reliably result positive; e.g., ~100,000 times more virus is needed to yield a positive result by the LumiraDx SARS-CoV-2 Ag Test than the PerkinElmer New Coronavirus Nucleic Acid Detection Kit [42, 43]. Additionally, SARS-CoV-2 can infect different upper-respiratory compartments, so numerous specimen types are used to detect infection (e.g., anterior-nares nasal swab, mid-turbinate nasal swab, nasopharyngeal swab, oropharyngeal swab, tonsillar swab, buccal swab, lingual swab, gingival crevicular fluid, saliva). The rise and fall of viral loads in each specimen type throughout infection affects whether SARS-CoV-2 is detectable in that specimen type at the time of testing. A diagnostic test successfully detects infection when the viral load in the tested specimen type is above the limit of detection (LOD) of the test.

In our recent analysis [44] of viral loads from three specimen types (anterior-nares swab, oropharyngeal swab, and saliva) prospectively collected daily before or at the incidence of infection with the Omicron variant, we observed that longitudinal viral-load timecourses in different specimen types from the same person often exhibit extreme differences and do not correlate. Further, most people in that study [44] and our prior study of ancestral variants [45] had delayed accumulation of virus in nasal swabs compared with oral specimens. A delayed rise in nasal-swab viral loads has been observed in many studies [46–49], including among participants in a SARS-CoV-2 human challenge study who received intra-nasal inoculation [50]. We [51] and others [44, 47, 49, 52, 53] found that this delayed rise in nasal viral loads, in combination with the high levels of virus required for detection by tests with low analytical sensitivity, leads to delayed detection of infected and infectious individuals by nasal-swab rapid antigen tests. Non-nasal upper respiratory specimen types and/or tests with high-analytical-sensitivity could detect these individuals earlier in the infection [44].

In this study, we investigated whether previously observed viral-load patterns that affect the clinical sensitivity of low analytical sensitivity rapid COVID-19 tests have implications

for household transmission. We specifically tested whether the type of test (rapid nasal-swab vs all other COVID-19 tests) used to first identify household index cases was correlated with higher rates of transmission to household contacts. Data were collected from a 2-year COVID-19 household transmission study in Southern California. We applied a two-level random intercept model, clustering by household and controlling for potential confounders [54] to assess the relationship between the use of a nasal-swab rapid COVID-19 tests to first identify the household index case, and subsequent transmission to household contacts (**Fig 1**).

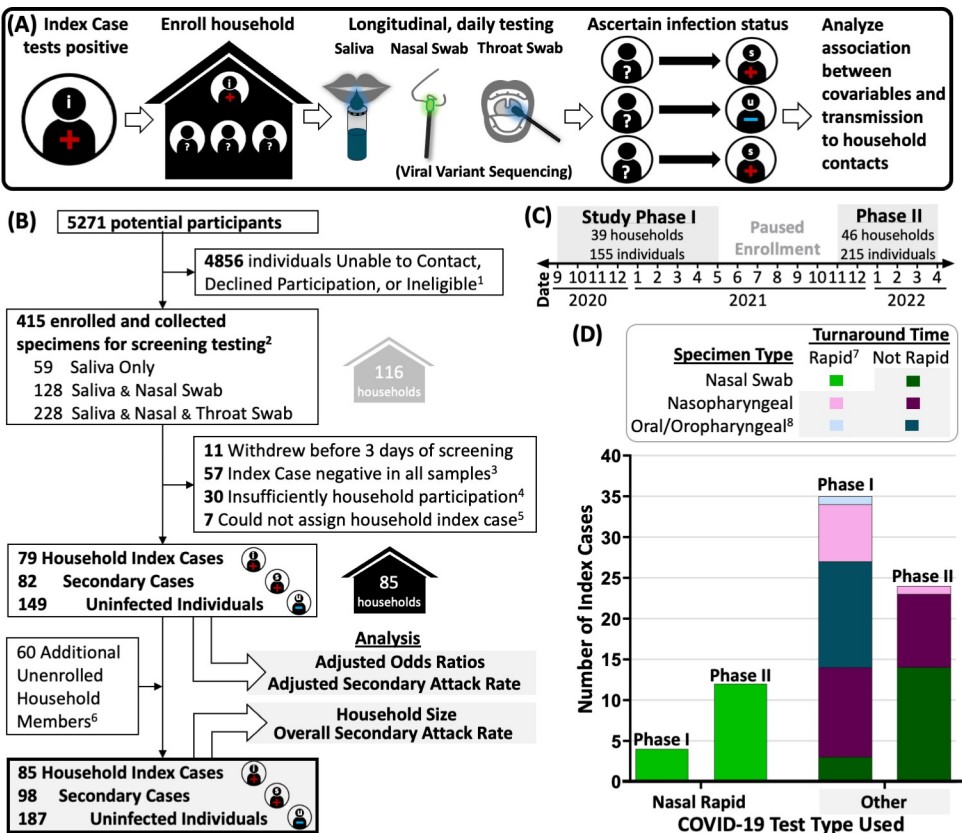

**Fig 1. Overview of study design and analysis. (A)** Study design beginning with recording the COVID-19 test type first used to identify index cases at study enrollment, enrollment of household contacts for daily, high-analytical-sensitivity laboratory screening, and analysis of potential factors modulating transmission. **(B)** CONSORT diagram for study enrollment. **(C)** Timeline of participant enrollment in study Phase I (September 2020—June 2021) and Phase II (November 2021—March 2022). Date is listed as numeric month over year. **(D)** Breakdown of self-reported COVID-19 test types (specimen type, and rapid or not) utilized to first identify household index cases. Test type was not reported by 10 of 85 index cases. 1. Individuals were ineligible for enrollment if they resided outside study jurisdiction, lived alone, or were more than 7 days from positive result or symptom onset. 2. Participants in Phase I collected either saliva only, or paired saliva and nasal swabs; participants in Phase II collected paired saliva, nasal swabs, and throat swabs. 3. Households were considered not at risk if no member including the suspected index case had detectable SARS-CoV-2 in any sample tested upon enrollment. 4. Households in which a majority of unenrolled household members were considered to have insufficient information. 5. Households in which a single household index case could not be assigned. 6. Information about unenrolled household members was reported by enrolled participants. 7. Test type was defined as 'Rapid' if the participant reported receiving results either within an hour or on the same day as the specimen was collected. Longer turnaround times were classified as 'Not Rapid' tests. 8. Oral/oropharyngeal specimen type category included participants who self-reported that saliva, buccal swabs, or oropharyngeal swabs were collected for testing.

## Materials and methods

### Participant enrollment and metadata

We conducted a case-ascertained COVID-19 household transmission observational cohort study in Southern California in two phases: between September 2020 and June 2021 [45, 55], prior to the predominance of the Delta variant [56], and between November 2021 and March 2022 [44], during the emergence and subsequent predominance of the Omicron variant [56] (**S1 Table in S1 File**). The study was approved by the California Institute of Technology IRB (protocol #20–1026). Individuals aged 6 and older were eligible for participation. Participants aged 8 years and older provided written informed consent, and all minors additionally provided verbal assent accompanied by written parental permission. Only the study coordinators, study administrator, and study PI had access to identifying information; the rest of the investigators were blinded to participant identity (see **S1 File**).

Upon enrollment, participants completed a questionnaire to provide information about demographics (see **S1 File**). At the conclusion of their participation, participants were asked to complete another questionnaire to report any SARS-CoV-2 test results from outside of the study, updated infection status of each household member (including those unenrolled), and infection-control practices performed.

### Laboratory screening testing

Self-collected specimens (saliva, anterior nares swabs, oropharyngeal swabs, **Fig 1A and 1B**) from participants at home underwent laboratory testing for SARS-CoV-2 infection by a high analytical sensitivity RT-qPCR test validated to have a limit of detection at or below 1,000 copies/mL for all specimen types in the study, as previously described (**S1 File**) [44, 45, 55]. Participants reported COVID-19-like symptoms at each specimen collection timepoint. At least one specimen from most households underwent viral sequencing as previously described [44, 45], to ascertain the infecting SARS-CoV-2 variant of household members. For one household enrolled in early December 2022, sequencing was not performed but Delta variant was inferred based on the dominating variants circulating at the time [56] and for 5 households enrolled after mid-January 2022, sequencing was not performed, but Omicron variant was inferred based on local predominance [56].

### Statistical analyses

We utilized the questionnaire data and laboratory testing data to investigate SARS-CoV-2 transmission within households. Households were included in this analysis if laboratory testing confirmed at least one household member was acutely infected with SARS-CoV-2 and more than a third of reported household members were enrolled in the study. Three households were excluded because they withdrew before three days of screening, 22 households were excluded because all members were negative for SARS-CoV-2 in all tested specimens, five households were excluded because of insufficient information about unenrolled household contacts, and one household was excluded because of inability to determine index case (**Fig 1B**). See **S1 File** for details.

For each household, an index case was defined as the first member of the household (enrolled in the study or not) to test positive for SARS-CoV-2 infection, usually prior to enrollment. In one case where multiple members had the same first test date, the member with earlier self-reported onset of symptoms was considered the household index case. In five cases where symptom onset of household members was within 1 day of each other, we defined the index case as the individual with a known exposure to a non-household contact with

laboratory-confirmed SARS-CoV-2 infection. In three cases with similar timing of exposure to infected, non-household contacts, the index case was defined as the individual whose viral load peaked first. All other members of the household who tested positive for SARS-CoV-2 prior to or during household enrollment in the study were considered secondary cases. Household members who never tested positive for SARS-CoV-2 prior to or while the household was enrolled in the study were considered uninfected. 143 of 149 (96%) participants classified as uninfected were enrolled and screened for at least 5 days; most (53%) were enrolled for at least 9 days.

The test type of the household index case was interpreted as a "nasal-swab rapid test" when the household index case self-reported "shallow nasal swab (anterior nares or mid turbinate nasal swab)" as the specimen type and a result turnaround time of "within an hour" or "same day." Participants were not asked to report the specific test name, laboratory platform, or viral target (e.g., molecular, antigen), due to concerns that laypersons would not be aware of these terms (especially if the test was run by a clinic rather than direct-to-consumer). However, rapid tests (both antigen and molecular) have characteristically low analytical sensitivity because they forego the time-consuming and technically challenging extraction steps to purify and concentrate viral targets. Because our hypothesis was related to low-analytical-sensitivity rapid tests performed on specimens from nasal swabs, we simply distinguish rapid tests from those with longer turnaround times and presumably higher analytical sensitivity (**Fig 1D**).

We calculated unadjusted odds ratios (ORs) for *a priori* confounders [57], infection-control practices, the use of nasal-swab rapid tests by index cases, and the risk of SARS-CoV-2 transmission to household contacts using mixed-effect logistic regression (**Fig 2**, **S2 Table in S1 File**). We also used a two-level mixed-effects logistic regression model with random intercepts by household to account for clustering of individuals within households and including all covariables to estimate adjusted odds ratios (aOR) (**Fig 2, S3 Table in S1 File**). This type of model [58] was chosen to estimate the effects of predictors at both individual and household levels. The model adjusted for a sufficient set of the following potentially confounding variables: household size [10–12], age [6, 15–17], race/ethnicity [11, 13], and vaccination status [19–23]. Observations with missing data were omitted from respective analyses. We also accounted for infecting SARS-CoV-2 viral variant [19–21, 29, 33, 34], either Omicron variant or ancestral variants. The study's second enrollment period occurred during increasing Omicron variant predominance. Thus, few participants in this study had Delta variant exposure, and analyses stratified by Delta variant infection would be insufficiently powered.

We used this model to assess the effect of household prevention practices and the COVID-19 test type used to first identify the household index case. An aOR >1.0 was associated with increased likelihood of household transmission, and deemed statistically significant if its associated *P*-value was ≤0.05 by Wald and likelihood ratio tests.

Predictive margins based on the results of the regression models were used to estimate unadjusted and adjusted secondary attack rates (SAR and aSAR). Binomial confidence intervals (CIs) were calculated as recommended by the Clinical Laboratory Standards Institute EP12-A guidance [59]. Differences among SARs and aSARs were assessed across strata [60].

We separately assessed the conditional direct effects of viral variant and test type used to identify the household index case by modifying the model with or without each of these covariables (**Fig 3**). Calculations were performed in STATA/BE 17.0.

## Results

We analyzed data from 370 individuals (enrolled participants and unenrolled household contacts reported by participants) of which 85 were defined as household index cases (**Fig 4**). Among index cases, nasal-swab rapid test use more than tripled from the first to second study

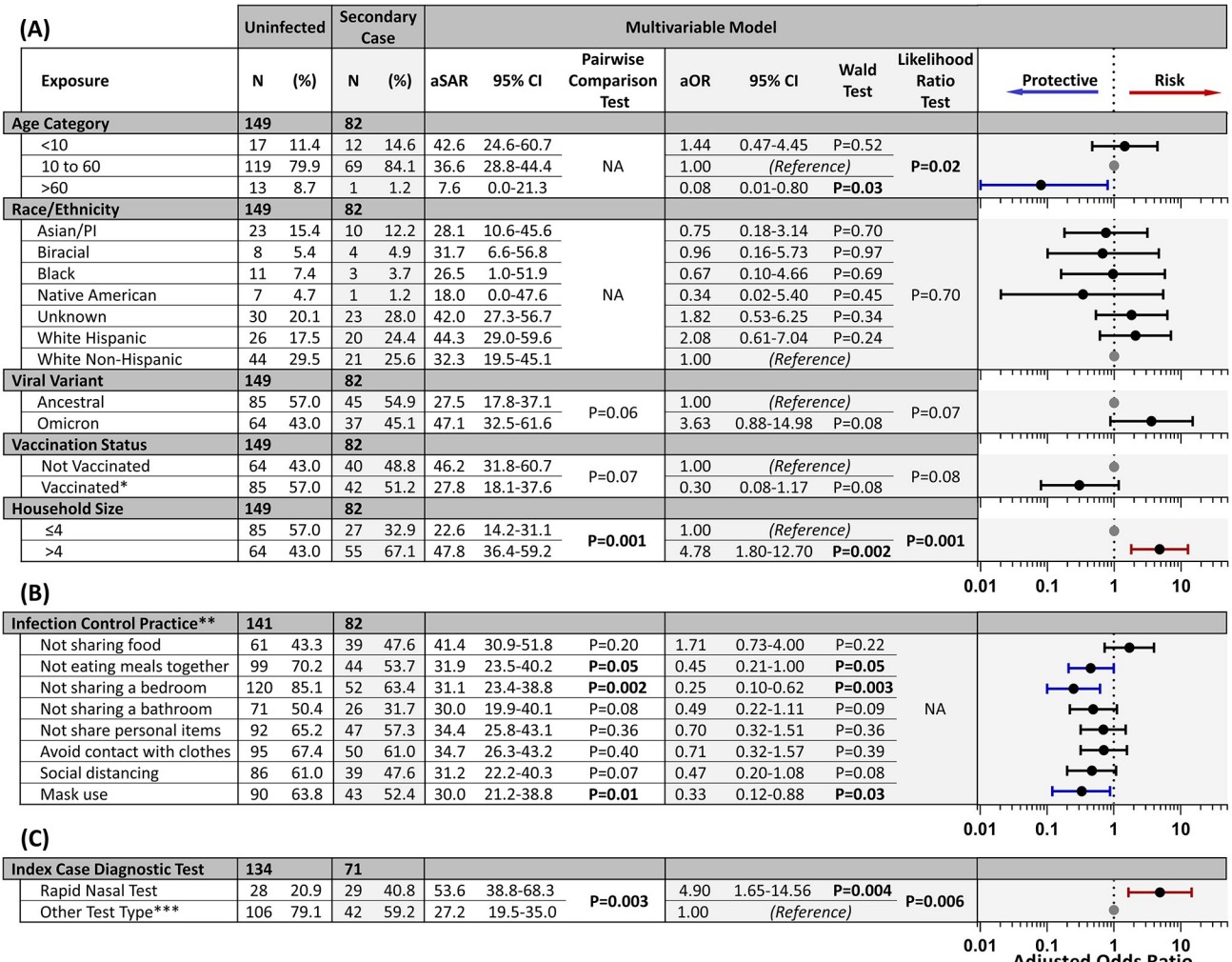

**Fig 2. Results of modeled risk of transmission to household contacts.** Counts (N) of enrolled individuals who did not become infected during enrollment (uninfected) or became infected after the index case (secondary case) are provided for each covariable included in the multivariable model (**Fig 1C**). The adjusted secondary attack rate (aSAR) and adjusted odds ratio (aOR) point estimates with 95% confidence intervals from multivariable analysis are listed for each covariable and visualized to the right. Results of univariable analysis are provided in **S2 Table in S1 File**. The Wald test *P*-values for the analyses likelihood ratio test is shown. Covariates with an aOR 95% CI >1 are shown in red, and those <1 are shown in blue. Reference groups are shown as a grey point. **(A)** Data for the five covariables included in the sufficient set. **(B)** Covariables related to infection-control practices controlling for the sufficient set. The aOR represents the conditional effect of the covariable in the model. **(C)** Association between COVID-19 test type used to identify the household index case, and subsequent transmission to household contacts. Unenrolled household index cases' test type was unknown, resulting in a lower total count for this category. *Vaccinated is defined as having received at least one dose of a COVID-19 vaccine at least 7 days prior to enrollment. **Participants were asked to respond whether or not they performed each action during interactions data coded. Data on infection control practices was not available for some participants. Observations with missing data were omitted, resulting in a lower total count for this category of covariables. ***Analysis by Other Test Type subgroups is shown in **S3 Table in S1 File**. Analysis by alternate age categories is shown in **S4 Table in S1 File**.

phase (**Fig 1D**). Only 3 of 16 index cases first identified by a rapid nasal-swab rapid tests had a prior negative rapid nasal-test within three days of their positive result, suggesting repeat rapid nasal testing [61]. Across both study phases, we observed an overall, unadjusted SAR of 34.4% (95% CI 28.9%–40.2%, 98 of 285 household contacts) in this population.

Without accounting for index case testing, we observed several covariables associated with SARS-CoV-2 transmission in households (**Fig 2**). Household size greater than four members was associated with nearly a 5-fold increase in the odds of infection (aOR = 4.78, 95% CI 1.80–

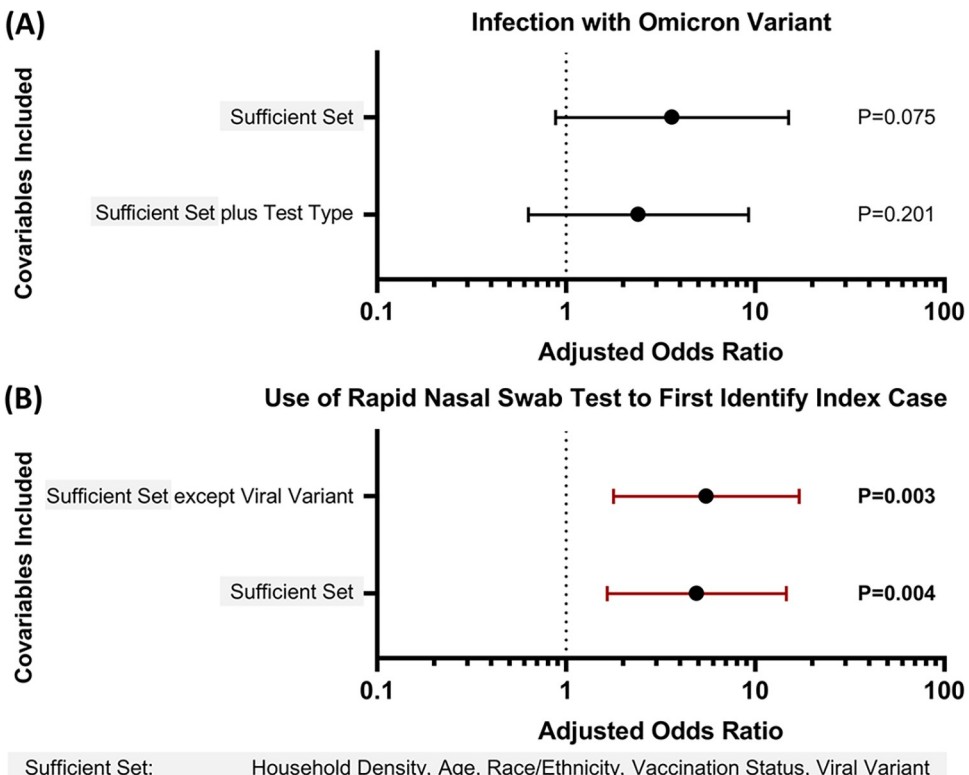

**Fig 3. Effect size interactions of COVID-19 test type and viral variant on transmission to household contacts. (A)** The adjusted odds ratios (aOR) for infection with the Omicron variant (with ancestral SARS-CoV-2 variants as reference). Analysis was performed while controlling for the sufficient set of covariables in the model (grey box), as well as when additionally controlling for whether the index case was first identified using a nasal-swab rapid test or other COVID-19 test type. **(B)** The aOR for the use of nasal-swab rapid tests to first identify index cases, as opposed to other COVID-19 test type. Analysis was performed while controlling for the sufficient set of covariables in the model (shown in grey box), and with all covariables in the sufficient set except for viral variant. Wald test *P*-values are shown for each estimate of effect size. All error bars are 95% CI. Vertical dotted black line indicates an aOR of 1.0.

12.70). Whether a household contact had received at least one dose of a COVID-19 vaccine was found to reduce the odds of infection by 70% (aOR = 0.30, 95% CI 0.08–1.17). Most infection-control practices were associated with reduced risk, such as not sharing a bedroom with (aOR = 0.25, 95% CI 0.10–0.62) and wearing masks around (aOR = 0.33, 95% CI 0.12–0.88) infected individuals.

We observed that infection with the Omicron variant was associated with greater transmission than ancestral variants. Increased transmissibility of the Omicron variant compared to ancestral variants was observed in our study by both aOR (3.64, 95% CI 0.88–15.07), as well as aSAR stratified by whether the index case was infected with the Omicron variant (46.9%, 95% CI 32.3%-61.6%) or an ancestral variant (27.3%, 95% CI 17.7%-36.9%). Increased transmissibility of the Omicron variant was not observed in the univariable model (**S2 Table in S1 File**), likely because this model does not correct for a compensating, protective effect of vaccination, which was more prevalent among individuals from households infected with the Omicron variant (76.7%) than ancestral variants (17.5%, **S1 Table in S1 File**).

Identification of index cases by nasal-swab rapid tests was associated with higher transmission to household contacts than other test types, both when aggregated (**Fig 2C**) and for all other test type subgroups (**S3 Table in S1 File**), and in both univariable (OR = 2.64, 95% CI 1.41–4.95, *P* = 0.003, **S2 Table in S1 File**) and multivariable models (aOR = 4.93, 95% CI 1.65–

| | | Index Case N= 85 | | Secondary Case N= 98 | | Uninfected N= 187 | | Total N= 370 | |
|---|---|---|---|---|---|---|---|---|---|
| **Self-Reported Gender Identity*** | | n | (%) | n | (%) | n | (%) | n | (%) |
| | Male | 31 | 36.5 | 50 | 51.0 | 91 | 48.7 | 172 | 46.5 |
| | Female | 52 | 61.2 | 45 | 45.9 | 96 | 51.3 | 193 | 52.2 |
| | Non-Binary/Other | 0 | 0.0 | 0 | 0.0 | 0 | 0.0 | 0 | 0.0 |
| | Unknown | 2 | 2.4 | 3 | 3.1 | 0 | 0.0 | 5 | 1.4 |
| **Age Category (years)** | | n | (%) | n | (%) | n | (%) | n | (%) |
| | <10 | 4 | 4.7 | 19 | 19.4 | 33 | 17.6 | 56 | 15.1 |
| | 10 to 60 | 78 | 91.8 | 75 | 76.5 | 133 | 71.1 | 286 | 77.3 |
| | >60 | 2 | 2.4 | 2 | 2.0 | 21 | 11.2 | 25 | 6.8 |
| | Unknown | 1 | 1.2 | 2 | 2.0 | 0 | 0.0 | 3 | 0.8 |
| **Self-Reported Race/Ethnicity**** | | n | (%) | n | (%) | n | (%) | n | (%) |
| | Asian or Pacific Islander | 14 | 16.5 | 10 | 10.2 | 23 | 12.3 | 47 | 12.7 |
| | Biracial | 4 | 4.7 | 4 | 4.1 | 8 | 4.3 | 16 | 4.3 |
| | Black/African American | 2 | 2.4 | 3 | 3.1 | 11 | 5.9 | 16 | 4.3 |
| | Native American/Alaska Native | 4 | 4.7 | 1 | 1.0 | 7 | 3.7 | 12 | 3.2 |
| | Unknown | 20 | 23.5 | 39 | 39.8 | 68 | 36.4 | 127 | 34.3 |
| | White, Hispanic | 22 | 25.9 | 20 | 20.4 | 26 | 13.9 | 68 | 18.4 |
| | White, Non-Hispanic | 19 | 22.4 | 21 | 21.4 | 44 | 23.5 | 84 | 22.7 |
| **Vaccination Status**** | | n | (%) | n | (%) | n | (%) | n | (%) |
| | Unvaccinated | 43 | 50.6 | 40 | 40.8 | 65 | 34.8 | 148 | 40.0 |
| | Partial | 0 | 0.0 | 2 | 2.0 | 3 | 1.6 | 5 | 1.4 |
| | Complete | 19 | 22.4 | 23 | 23.5 | 36 | 19.3 | 78 | 21.1 |
| | Boosted | 17 | 20.0 | 17 | 17.3 | 45 | 24.1 | 79 | 21.4 |
| | Unknown | 6 | 7.1 | 16 | 16.3 | 38 | 20.3 | 60 | 16.2 |
| **Household Viral Variant** | | n | (%) | n | (%) | n | (%) | n | (%) |
| | Ancestral | 39 | 45.9 | 37 | 37.8 | 79 | 42.2 | 155 | 41.9 |
| | Delta | 12 | 14.1 | 14 | 14.3 | 22 | 11.8 | 48 | 13.0 |
| | Omicron | 28 | 32.9 | 45 | 45.9 | 64 | 34.2 | 137 | 37.0 |
| | Unknown | 6 | 7.1 | 2 | 2.0 | 22 | 11.8 | 30 | 8.1 |
| **Smoking/Vaping History History** | | n | (%) | n | (%) | n | (%) | n | (%) |
| | Never | 55 | 64.7 | 56 | 57.1 | 104 | 55.6 | 215 | 58.1 |
| | Former | 16 | 18.8 | 14 | 14.3 | 24 | 12.8 | 54 | 14.6 |
| | Current | 7 | 8.2 | 4 | 4.1 | 6 | 3.2 | 17 | 4.6 |
| | Unknown | 7 | 8.2 | 24 | 24.5 | 53 | 28.3 | 84 | 22.7 |

**Fig 4. Demographics, COVID-19 vaccination status, viral variant, and smoking history of the 85-household cohort used for analyses.** *Both sex assigned at birth and current gender identity were self-reported by participants. One participant reported male assignment at birth and current gender identity of woman. Reported gender is listed. **63 individuals currently listed as 'Unknown' did not select a race category but wrote-in "Latino"/"Latina"/"Latinx". ***Participants reported date and manufacturer of each vaccine dose received; vaccination status was defined only by doses received at least 7 days prior to enrollment in the study. Unvaccinated was defined as having received no COVID-19 vaccine doses. Partial vaccination was defined as receiving one dose of a multiple-dose series (e.g., Pfizer-BioNTech, Moderna). Complete vaccination was defined as receiving all doses of an initial COVID-19 vaccine series. Boosted was defined as the participant receiving any dose beyond an initial COVID-19 vaccine series. Vaccination and viral variant distributions varied by Study Phase; demographics by Study Phase are shown in **S1 Table in S1 File**.

14.69, $P$ = 0.004, **Fig 2**). The multivariable model suggests that nasal-swab rapid test use by index cases increased the odds of transmission relative to other test types by almost five-fold (though both smaller and larger increases are also compatible with the data). Index cases who used nasal-swab rapid tests also had a higher aSAR of 53.5% (95% CI 38.7%–68.3%) compared to other test types (27.0%, 95% CI 19.3%–34.8%).

Because the use of nasal-swab rapid test use has increased in parallel with SARS-CoV-2 variants shown to have increased transmissibility, we examined the relationship of these two covariables on risk of transmission to household contacts. The use of a nasal-swab rapid test to identify the index case was associated with a similar increased risk of transmission to household contacts) as infection with the Omicron variant (**Fig 3**). Introducing adjustment in the model for nasal-swab rapid test use by the index case decreased the aOR for infection with the Omicron variant from 3.63 (95% CI 0.88–15.0) to 2.40 (95% CI 0.63–9.22) (**Fig 3A**). The aOR of rapid nasal-swab test use also decreased from 5.50 (95% CI 1.78–17.04) to 4.90 (95% CI 1.65–14.59) without or with adjustment for viral variant, but nasal-swab rapid tests remained associated with at least a 1.5-fold increase in the odds of household contact infection (**Fig 3B**). When the analysis was limited to the participants in households infected with the Omicron variant (**S5 Table in S1 File**), use of rapid nasal swab tests by index cases remained significantly associated with increased transmission to household contacts compared to other test types (aOR 15.89 95% CI 1.59.-158.41).

## Discussion

Household contacts of index cases who used nasal-swab rapid antigen COVID-19 tests for primary infection detection had an increased risk of becoming infected compared with household contacts of index cases who used other test types. Greater transmission of SARS-CoV-2 to household contacts by individuals first identified by nasal-swab rapid tests is supported mechanistically by studies of SARS-CoV-2 viral load and nasal swab rapid test performance. First, a gradual rise in viral loads, as we [44, 45, 51, 62] and others [53, 63–65] have observed, often creates a several-day delay between when an individual likely becomes infectious and when viral loads reach levels detectable by low-analytical-sensitivity, rapid tests. Second, a delay in the rise of nasal viral loads relative to oral specimen types, as we [44, 45, 51] and others [46, 50] have observed, renders nasal-swab rapid tests less able to detect individuals during the early phase of the infection [47, 51]. During this early period of low nasal viral loads, we [44, 51] and others [47] find that individuals exhibited high, presumably infectious viral loads in oral specimens. Relatedly, among data from a SARS-CoV-2 human challenge study [50], we see that the majority of infected participants had replication-competent virus present in throat swabs at least one day prior to nasal-swabs. Therefore, nasal-swab rapid tests may only yield positive results after exposure and transmission to contacts has occurred. These results together suggest that nasal-swab rapid tests are not as effective at identifying index cases to limit subsequent transmission as other test types.

Several additional findings from our model and dataset were consistent with prior studies. Household size was a significant risk factor for household transmission [9–12], whereas vaccination [8, 19–23] and infection-control practices [14, 19, 24–26] were protective. The overall SAR (34.4%) we observed was similar to what others have reported [5, 12, 19, 32, 66, 67]. Relatedly, in one of those studies [5], household transmission was monitored by daily high-analytical-sensitivity screening testing and the SAR calculated using only nasal-swab test data was lower than when both saliva and nasal-swab test data were used, which supports that even high-analytical-sensitivity nasal-swab testing may miss some infected individuals, and that the specimen type used for evaluation can impact estimates of transmission.

We also observed, as other epidemiological studies have [8, 19, 20, 32–34], that infection with the Omicron variant was associated with increased transmission compared with ancestral viral variants. However, the use of rapid nasal-swab tests (as opposed to other test types) to detect index cases had a similar conditional direct effect on transmission to household contacts as infection with the Omicron variant. Because the effect size of the Omicron variant

association with transmission to household contacts decreased when controlling for nasal-swab rapid test use in our study, we speculate that a portion of the increased transmissibility attributed to the Omicron variant in published epidemiological studies may be partially attributable to the increased use of rapid nasal-swab tests in the U.S. that coincided with the predominance of this variant [10, 68]. Although our results do not invalidate studies that conclude an increased transmissibility of the Omicron variant, they emphasize the potential impact of COVID-19 test type on estimates of transmissibility from epidemiological data.

Our findings are subject to limitations. First, vaccination status, demographic information, and infection-control practices are self-reported and may be subject to recall bias. Second, although questionnaires were written in simple terms (e.g. "shallow nasal swab" and "deep nasal swab"), participants could have misinterpreted test type. Third, age, gender, and infection status of each unenrolled household member was independently reported by each enrolled household member, which could lead to inaccurate reporting. Fourth, our potential misclassification of which household member was the index case may impact the analysis [54], although in almost all (79 of 85) households, the index case was confirmed by timing of self-reported positive tests. Fifth, in our transmission model, we did not analyze ordinal levels of contact among household members (all household members were assumed to have equal contact). Instead, mitigating factors, including infection-control practices, were assessed for protective effects against transmission. Sixth, it is possible that high-analytical-sensitivity tests could have turnaround times which we classify as rapid. However, such misclassification would bias toward the null. Finally, evidence suggests [53, 69] and the CDC [61] recommends repeating rapid antigen tests over several days to improve clinical sensitivity. Although some index cases reported a negative test result in the days prior to their first positive result, most participants in our study did not use repeated rapid testing.

## Conclusion

Rapid COVID-19 tests, such as antigen tests, are less expensive, portable, and offer faster results than high-analytical-sensitivity molecular tests. However, results from this observational study suggest that the use of nasal-swab rapid COVID-19 tests to first identify infection does not limit household transmission as effectively as other test types. The use of tests with low analytical sensitivity by an infected individual can have two effects on transmission: (i) a true-positive result can change behavior to increase infection-control practices in a timely manner, thus reducing transmission, or (ii) a false-negative result can result in a health certificate effect [70], where individuals falsely assume they are not infected/infectious and reduce precautions, thereby increasing transmission. While imperfect testing may be better than no testing, understanding the optimal use and limitations of rapid tests is important not only for SARS-CoV-2, but other pathogens for which timely infection control and/or early treatment is critical.

## Supporting information

**S1 File.** Containing S1 Table (Participant Demographics by [A] Study Phase and [B] Infecting SARS-CoV-2 Variant), S2 Table (Univariable Model), S3 Table (Association of Test Type Subcategories with SARS-CoV-2 Transmission Among Household Contacts), S4 Table (Results of Modeled Risk of Transmission to Household Contacts with Alternative Age Grouping), S5 Table (Results of Modeled Risk of Transmission to Household Contacts of Index Cases Infected with Omicron Variant), Supplemental Methods, and Supplemental References. (PDF)

## Acknowledgments

We thank the University of California, Los Angeles, Office of Advanced Research Computing, Statistical Methods and Data Analytics Group for recommendations on statistical methodology and implementation, and Dr. Andy Lin for guidance designing the analysis and feedback on the manuscript.

## Author Contributions

**Conceptualization:** Jenny Ji, Alexander Viloria Winnett, Ying-Ying Goh, Matt Feaster, Rustem F. Ismagilov.

**Data curation:** Jenny Ji, Alexander Viloria Winnett, Natasha Shelby.

**Formal analysis:** Jenny Ji, Alexander Viloria Winnett, Matt Feaster.

**Funding acquisition:** Jenny Ji, Alexander Viloria Winnett, Rustem F. Ismagilov.

**Investigation:** Jenny Ji, Alexander Viloria Winnett, Natasha Shelby, Jessica A. Reyes, Noah W. Schlenker, Hannah Davich, Saharai Caldera, Colten Tognazzini.

**Project administration:** Rustem F. Ismagilov.

**Resources:** Colten Tognazzini, Ying-Ying Goh, Matt Feaster.

**Supervision:** Natasha Shelby, Rustem F. Ismagilov.

**Validation:** Jenny Ji, Alexander Viloria Winnett, Natasha Shelby, Jessica A. Reyes, Noah W. Schlenker, Hannah Davich, Saharai Caldera.

**Visualization:** Jenny Ji, Alexander Viloria Winnett, Natasha Shelby.

**Writing – original draft:** Jenny Ji, Alexander Viloria Winnett.

**Writing – review & editing:** Jenny Ji, Alexander Viloria Winnett, Natasha Shelby, Matt Feaster, Rustem F. Ismagilov.

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
