## [Decision Letter · Decision Letter 0]

17 Jul 2023

PONE-D-23-09614Index Cases First Identified by Nasal-Swab Rapid COVID-19 Tests Had More Transmission to Household Contacts Than Cases Identified by Other Test TypesPLOS ONE

Dear Dr. Ismagilov,

Thank you for submitting your manuscript to PLOS ONE. After careful consideration, we feel that it has merit but does not fully meet PLOS ONE’s publication criteria as it currently stands. Therefore, we invite you to submit a revised version of the manuscript that addresses the points raised during the review process. Please carefully revise your manuscript, particularly with reference to the Reviewer's comments regarding potential differences between the Phase 1 and Phase 2 studies, as well as comments from both reviewers regarding the effects observed based on the different variants (e.g. Delta).

We look forward to receiving your revised manuscript.

Kind regards,

Paul V. Licciardi

Academic Editor

PLOS ONE

Journal Requirements:

"This study is based on research funded in part by the Bill & Melinda Gates Foundation (INV-023124). The findings and conclusions contained within are those of the authors and do not necessarily reflect positions or policies of the Bill & Melinda Gates Foundation. This study was also funded in part by grants from the Ronald and Maxine Linde Center for New Initiatives at the California Institute of Technology (to RFI), a grant from the Jacobs Institute for Molecular Engineering for Medicine at the California Institute of Technology (to RFI), a DGSOM Geffen Fellowship at the University of California, Los Angeles (to AVW), and the John Stauffer Charitable Trust SURF Fellowship at the California Institute of Technology (to JJ)."

"This study is based on research funded in part by the Bill & Melinda Gates Foundation (INV-023124). The findings and conclusions contained within are those of the authors and do not necessarily reflect positions or policies of the Bill & Melinda Gates Foundation. This study was also funded in part by grants from the Ronald and Maxine Linde Center for New Initiatives at the California Institute of Technology (to RFI), a grant from the Jacobs Institute for Molecular Engineering for Medicine at the California Institute of Technology (to RFI), a DGSOM Geffen Fellowship at the University of California, Los Angeles (to AVW), and the John Stauffer Charitable Trust SURF Fellowship at the California Institute of Technology (to JJ)."

"I have read the journal's policy and the authors of this manuscript have the following competing interests: R.F.I. is a cofounder, consultant, and a director and has stock ownership of Talis Biomedical Corp. All other co-authors report no competing interests."

Reviewers' comments:

Reviewer's Responses to Questions

**Comments to the Author**

1. Is the manuscript technically sound, and do the data support the conclusions?

Reviewer #1: Yes

Reviewer #2: Partly

2. Has the statistical analysis been performed appropriately and rigorously? 

Reviewer #1: Yes

Reviewer #2: I Don't Know

3. Have the authors made all data underlying the findings in their manuscript fully available?

Reviewer #1: Yes

Reviewer #2: Yes

4. Is the manuscript presented in an intelligible fashion and written in standard English?

Reviewer #1: Yes

Reviewer #2: Yes

5. Review Comments to the Author

Reviewer #1: Although COVID-19 infection currently is not a public health emergency, the lessons learnt can be used for future pandemic. The deep understanding of COVID-19 infection will support the response to other respiratory virus. Household transmission is a challenge to control the infection. The authors presented a relevant study to the understand the role of rapid tests to control the transmission of COVID-19 among the household members.

1. Delta and Omicron variants were concern to public health systems due to high transmission rate. Why the authors only conducted the analysis for Omicron variant. I recommend to do the same for Delta variant.

2. Several rapid tests, with different specificity and sensitivity, were developed for COVID-19 diagnosis. Assuming the variation on diagnostic performance, which rapid test was considered for this study? The test sensitivity is not a confounding?

3. The authors described that the study participants collected not only nasal swabs. Were these specimens for rapid testing or RT-PCR? Include a statement in the methods to clarify this.

Was the oropharyngeal specimen self-collected?

4. What was the minimum age of children included in the study? 6 or 8 years old? The age on main manuscript is different of the supplementary material?

5. Line 176, Fig2A, how the unenrolled participants defined. Why did you include the unenrolled participants?

Reviewer #2: This manuscript described the SARS-CoV-2 transmission within 85 households and compared the transmission rate between nasal-swab rapid test and “other test types” grouped into other 6 types (nasal and not rapid, nasopharyngeal rapid, nasopharyngeal and not rapid, oral rapid and oral not rapid).

While the author provided rationale and prior evidence to support carrying out these analyses, the title/conclusion do not match the data presented (other factors that may contribute to the observed data) and there are major queries to the analyses, particularly when combining Phase 1 and Phase 2 participants. Please see below:

Major comments/queries:

- As mentioned by the authors, the type of samples (saliva, nasal, nasopharyngeal) and the type of test have varied sensitivities and specificities Furthermore, the sample medium and some of the sample types collected in Phase 1 and 2 are different. Please provide information regarding the type of test (RT-qPCR, saliva rapid test), and do you see a similar results if you compare nasal vs saliva vs nasopharyngeal and/or RT-qPCR vs rapid test regardless of sample types, which gene the test detects)? Do you expect the sample medium to influence the test sensitivity?

- As acknowledged by the author, the variant type is an important factor in transmission. The author concluded that rapid nasal swab increased transmission, if you compare nasal-swab rapid test and “other test types” within the Phase 1 (ancestral strain) and within Phase 2( omicron), do you see the same results?

- What is the rationale for grouping individuals aged 10-60 years in a group? Host response and transmission in children or young adolescents are different from adults. Do you see the same results if the participants are stratified into different age groups, e.g. <12 years, 12-18 years, 19-60 years and >60 years.

- Figure 3b- do you include infection control practices as a covariable for analysis?

- Line 73: “we aimed to investigate whether test analytical sensitivity and differences in viral-load patterns among different specimen types may have implications for household transmission” – there are no investigation of viral load and test analytical sensitivity- the authors may have had shown this in previous publication on a subset of participants, but cannot assume this the same for all participants in this analyses?

Minor comments:

- Line 34. “Use of nasal-swab rapid COVID-19 tests for initial detection of infection and initiation of infection control may not limit transmission as well as other test types.” – this applies to household contacts.

- Line 187-188- should be in discussion

6. PLOS authors have the option to publish the peer review history of their article (what does this mean?). If published, this will include your full peer review and any attached files.

Reviewer #1: No

Reviewer #2: No

---

## [Author Response · Author response to Decision Letter 0]

14 Aug 2023

Please see uploaded cover letter for point-by-point response to Reviewers.

---

## [Decision Letter · Decision Letter 1]

4 Sep 2023

PONE-D-23-09614R1Index Cases first identified by nasal-swab rapid COVID-19 tests had more transmission to household contacts than Cases identified by other test typesPLOS ONE

Dear Dr. Ismagilov,

Thank you for submitting your manuscript to PLOS ONE. After careful consideration, we feel that it has merit but does not fully meet PLOS ONE’s publication criteria as it currently stands. Therefore, we invite you to submit a revised version of the manuscript that addresses the points raised during the review process.

We look forward to receiving your revised manuscript.

Kind regards,

Paul V. Licciardi

Academic Editor

PLOS ONE

Journal Requirements:

**Additional Editor Comments:**

Thankyou for addressing the Reviewer comments. Please respond to the additional Reviewer comments below before a final decision can be made.

Reviewers' comments:

Reviewer's Responses to Questions

**Comments to the Author**

1. If the authors have adequately addressed your comments raised in a previous round of review and you feel that this manuscript is now acceptable for publication, you may indicate that here to bypass the “Comments to the Author” section, enter your conflict of interest statement in the “Confidential to Editor” section, and submit your "Accept" recommendation.

Reviewer #1: (No Response)

Reviewer #2: (No Response)

2. Is the manuscript technically sound, and do the data support the conclusions?

Reviewer #1: Yes

Reviewer #2: Yes

3. Has the statistical analysis been performed appropriately and rigorously? 

Reviewer #1: Yes

Reviewer #2: Yes

4. Have the authors made all data underlying the findings in their manuscript fully available?

Reviewer #1: Yes

Reviewer #2: Yes

5. Is the manuscript presented in an intelligible fashion and written in standard English?

Reviewer #1: Yes

Reviewer #2: Yes

6. Review Comments to the Author

Reviewer #1: 1. Regarding the limited sample size to conduct the analysis for Delta variant of SARS-CoV-2, the authors should state this as study limitation. It is important because the authors referred that both delta and omicron variants were circulating.

Reviewer #2: Thank you for addressing the queries and appreciate the sub-analysis. I only have one minor comment in the abstract and discussion:

- line 35 and line 299: nasal-swab rapid test is less effective than other test, but that does not mean it does not limit transmission- please consider rewording to e.g. less effective than other test or something equivalent rather than 'no use' at all.

7. PLOS authors have the option to publish the peer review history of their article (what does this mean?). If published, this will include your full peer review and any attached files.

Reviewer #1: No

Reviewer #2: No

---

## [Editor Report · Decision Letter 2]

19 Sep 2023

Index Cases first identified by nasal-swab rapid COVID-19 tests had more transmission to household contacts than Cases identified by other test types

PONE-D-23-09614R2

Dear Dr. Ismagilov,

We’re pleased to inform you that your manuscript has been judged scientifically suitable for publication and will be formally accepted for publication once it meets all outstanding technical requirements.

Kind regards,

Paul V. Licciardi

Academic Editor

PLOS ONE
---

## [Editor Report · Acceptance letter]

25 Sep 2023

PONE-D-23-09614R2 

Index Cases first identified by nasal-swab rapid COVID-19 tests had more transmission to household contacts than Cases identified by other test types 

Dear Dr. Ismagilov:

I'm pleased to inform you that your manuscript has been deemed suitable for publication in PLOS ONE. Congratulations! Your manuscript is now with our production department. 

Kind regards, 

on behalf of

Dr. Paul V. Licciardi 

Academic Editor

PLOS ONE